# Use of glucocorticoids megadoses in SARS-CoV-2 infection in a spanish registry: SEMI-COVID-19

**Cristina Lavilla Olleros**[1◑*], **Cristina Ausín García**[1◑], **Alejandro David Bendala Estrada**[1◑], **Ana Muñoz**[2], **Philip Erick Wikman Jogersen**[3], **Ana Fernández Cruz**[4], **Vicente Giner Galvañ**[3], **Juan Antonio Vargas**[4], **José Miguel Seguí Ripoll**[3], **Manuel Rubio-Rivas**[5], **Rodrigo Miranda Godoy**[6], **Luis Mérida Rodrigo**[7], **Eva Fonseca Aizpuru**[8], **Francisco Arnalich Fernández**[9], **Arturo Artero**[10], **Jose Loureiro Amigo**[11], **Gema María García García**[12], **Luis Corral Gudino**[13], **Jose Jiménez Torres**[14], **José-Manuel Casas-Rojo**[15‡], **Jesús Millán Núñez-Cortés**[1‡], **On behalf of the SEMI-COVID-19 Network**[¶]

1 General Internal Medicine Department, Hospital Universitario Gregorio Marañón, Madrid, Spain, 2 General Internal Medicine Department, Infanta Cristina Hospital: Hospital Universitario Infanta 2 Cristina, Parla-Madrid, Spain, 3 General Internal Medicine Department, Hospital Universitario San Juan de Alicante, San Juan de Alicante-Alicante, Spain, 4 General Internal Medicine Department, Hospital Universitario Puerta de Hierro, Madrid, Spain, 5 General Internal Medicine Department, H. Univ. de Bellvitge, L'Hospitalet de Llobregat, Barcelona, Spain, 6 General Internal Medicine Department, 12th of October University Hospital: Hospital Universitario 12 de Octubre, Madrid, Spain, 7 General Internal Medicine Department, Hospital Costa del Sol, Málaga, Spain, 8 General Internal Medicine Department, Hospital de Cabueñes: Hospital de Cabuenes, Gijón, Spain, 9 General Internal Medicine Department, La Paz University Hospital: Hospital Universitario La Paz, Madrid, Spain, 10 General Internal Medicine Department, Hospital Universitario Dr Peset: Hospital Universitario Doctor Peset, Valencia, Spain, 11 General Internal Medicine Department, Hospital de Sant Joan Despí Moisès Broggi: Hospital de Sant Joan Despi Moises Broggi, Barcelona, Spain, 12 General Internal Medicine Department, University Hospital Complex Badajoz: Complejo Hospitalario Universitario de Badajoz, Badajoz, Spain, 13 Hospital Universitario Rio Ortega, Valladolid, Spain, 14 General Internal Medicine Department, Hospital Reina Sofía: Hospital Reina Sofia, Córdoba, Madrid, Spain, 15 General Internal Medicine Department, Infanta Cristina Hospital: Hospital Universitario Infanta Cristina, Parla-Madrid, Spain

◑ These authors contributed equally to this work.
‡ These authors also contributed equally to this work.
¶ Membership of the SEMI-COVID-19 Network is provided in the Acknowledgments.
* cristinalavillaolleros@gmail.com

**Data Availability Statement:** All relevant data are within the paper and its Supporting information files.

## Abstract

### Objective

To describe the impact of different doses of corticosteroids on the evolution of patients with COVID-19 pneumonia, based on the potential benefit of the non-genomic mechanism of these drugs at higher doses.

### Methods

Observational study using data collected from the SEMI-COVID-19 Registry. We evaluated the epidemiological, radiological and analytical scenario between patients treated with megadoses therapy of corticosteroids vs low-dose of corticosteroids and the development of complications. The primary endpoint was all-cause in-hospital mortality according to use of corticosteroids megadoses.

**Funding:** The authors received no specific funding for this work.

**Competing interests:** The authors have declared that no competing interests exist.

## Results

Of a total of 14,921 patients, corticosteroids were used in 5,262 (35.3%). Of them, 2,216 (46%) specifically received megadoses. Age was a factor that differed between those who received megadoses therapy versus those who did not in a significant manner (69 years [IQR 59–79] vs 73 years [IQR 61–83]; $p < .001$). Radiological and analytical findings showed a higher use of megadoses therapy among patients with an interstitial infiltrate and elevated inflammatory markers associated with COVID-19. In the univariate study it appears that steroid use is associated with increased mortality (OR 2.07 95% CI 1.91–2.24 p < .001) and megadose use with increased survival (OR 0.84 95% CI 0.75–0.96, p 0.011), but when adjusting for possible confounding factors, it is observed that the use of megadoses is also associated with higher mortality (OR 1.54, 95% CI 1.32–1.80; p < .001). There is no difference between megadoses and low-dose (p .298). Although, there are differences in the use of megadoses versus low-dose in terms of complications, mainly infectious, with fewer pneumonias and sepsis in the megadoses group (OR 0.82 95% CI 0.71–0.95; p < .001 and OR 0.80 95% CI 0.65–0.97; p < .001) respectively.

## Conclusion

There is no difference in mortality with megadoses versus low-dose, but there is a lower incidence of infectious complications with glucocorticoid megadoses.

## Introduction

On December 31st, 2019, Wuhan Municipal Health and Sanitation Commission (Hubei Province, China) reported 27 new cases of pneumonia of an unknown etiology to the World Health Organization. The agent causing this pneumonia has been identified as a virus of the *Coronaviridae* family, named SARS-CoV-2 (Severe Acute Respiratory Syndrome—Coronavirus 2) [1,2].

It has been postulated that in the development of the disease it is possible to distinguish a hyper-inflammatory phase, in which using glucocorticoids might play an essential role in preventing acute pulmonary distress syndrome (ARDS) [3–5].

Corticosteroids (CS) have been widely adopted, there are still questions on dosing, timing and duration of CS that have not been systematically studied at large.

The aim of this study is to describe the impact of different doses of corticosteroids on the evolution of patients with COVID-19 pneumonia.

## Literature search

A literature search was conducted using the MEDLINE database with the following search terms: "corticosteroids and COVID-19," "megadoses and SARS-CoV-2," and "immunomodulatory and COVID-19." The most up-to-date evidence and all information regarding use of corticosteroids in COVID-19 reported in English or Spanish were selected.

## Material and methods

This work is a multicenter, nationwide, observational study based on patient data obtained from the SEMI-COVID-19 Registry, an enterprise of the Spanish Society of Internal Medicine (SEMI, for its initials in Spanish) to advance knowledge of the patients infected with

SARS-CoV-2. The SEMI-COVID-19 Registry was approved by the Provincial Research Ethics Committee of Málaga (Spain).

## Study design and population

The registry is an anonymized online database of retrospective data on consecutive adult patients with COVID-19 hospitalized in internal medicine departments from 131 Spanish hospitals. The diagnosis was confirmed microbiologically by reverse transcription polymerase chain reaction (RT-PCR) testing of a nasopharyngeal or bronchoalveolar lavage sample. Exclusion criteria were subsequent admissions of the same patient and denial or withdrawal of informed consent. Patients were cared for at their attending physician's discretion, according to local protocols and their clinical judgment.

The registry includes data on more than 300 variables in categories such as:

- Sociodemographic and epidemiological data

- Personal medical and medication history

- Symptoms and physical examination findings upon admission

- Laboratory test results

- Radiological findings and their progress

- Pharmacological treatment and ventilatory support

- In-hospital complications and causes of death

More in-depth information on the registry and preliminary results are available in a previously published work [6].

## Study endpoints

The primary endpoint was all-cause in-hospital mortality according to use of corticosteroids megadoses, defined as > 150 mg of prednisone in 24h. The follow-up period was from admission to discharge or death, including early readmissions.

We analyzed the criteria for the use of megadoses, any relationship to epidemiological, clinical, laboratory, and radiologic parameters, and the development of complications depending on the use of megadoses of corticosteroids.

## Data analysis

We initially selected patients who received corticosteroids (5,262 out of a sample of 14,921). We further subdivided this population into two groups according to the amount of corticosteroids received: low-dose and megadoses. We defined megadoses therapy as the use of > 150 mg prednisone in 24h.

Continuous quantitative variables were tested for normal distribution using rates of skewness and kurtosis, Levene's test, or the Kolmogorov-Smirnov test, as appropriate. These variables were expressed as medians and interquartile range (IQR). Comparisons between groups were made using the Student's T-test, Mann-Whitney U test, Wilcoxon test, analysis of variance (ANOVA), or the Kruskal-Wallis test. Categorical variables were expressed as absolute values and percentages. Differences in proportions were analyzed using the chi-square test, McNemar's test, or Fisher's exact test, as appropriate.

Measures of association were expressed as odds ratio (OR) with 95% confidence intervals (95% CI). Statistical analysis was carried out using STATA software (v14.2). Statistical significance was established as $p < 0.005$.

We also used logistic regression to evaluate the relationship between use of megadoses and mortality. A multivariate analysis was carried out to adjust for confounding variables using clinically relevant, statistically significant variables ($p < 0.001$) identified in the previous analysis.

## Results

### Demographics, and clinical features

Demographics and comorbidities in patients with corticosteroids or megadose therapy are shown in Table 1. Age differed between those who received megadose therapy versus those who did not in a significant manner (69 years [IQR 59–79] vs 73 years [IQR 61–83]; $p <$ .001). There was a lower rate of megadose therapy among patients with dyslipidemia, arterial hypertension, heart and respiratory diseases. Regarding patients' previous treatment, a lower percentage of patients who were taking systemic corticosteroids therapy or other immunosuppressive received megadoses therapy.

### Laboratory and radiologic findings

Radiologic and laboratory findings showed a higher use of megadose therapy among patients with an interstitial infiltrate and elevated inflammatory markers associated with COVID-19, such as elevated lactate dehydrogenase and C-reactive protein, on admission. Full data are presented in Table 2.

### Other treatments

In Table 3 we studied the use of other treatments concomitantly for SARS-CoV-2 infection. There was a trend towards greater use of other immunomodulatory medications in those patients who also received corticosteroid megadoses.

### Complications and mortality

There are differences in the use of megadoses versus low-dose in terms of complications. This is reflected in Table 4. The risk of most complications was lower in the group of megadoses, especially those related to other infections (bacterial pneumonia OR 0.82, 95% CI 0.71–0.95; p .010 and sepsis 0.80 (0.65–0.97) OR 0.80, 95% CI 0.65–0.97; p .026). Although the risk was only higher in the case of stroke (OR 2.60, 95% CI 1.38–4.90; p .003, or venous thromboembolic disease (OR 1.72, 95% CI 1.26–2.33 p .001).

The analysis of outcome and mortality is shown in Table 5. We found no difference between ICU admission (14.4% low dose; 15% megadoses p .54) and average in hospital stay per days in both groups (12 days, IQR 7–18 low-dose; 12 days, IQR 8–19 megadoses. p .88).

Tables 6 and 7 show the relationship between steroid use and mortality. Patients were initially divided into two groups according to whether or not they received steroid therapy, and specifically the use of megadoses or low-dose of corticosteroids. In the univariate study it appears that steroid use is associated with increased mortality (OR 2.07 95% CI 1.91–2.24 p <0.001) and megadose use with increased survival (OR 0.84 95% CI 0.75–0.96, p 0.011), but when adjusting for possible confounding factors, it is observed that the use of megadoses is also associated with higher mortality (OR 1.54, 95% CI 1.32–1.80; p < .001). The low-dose

**Table 1. Demographics and comorbidities.**

| | N | Total population (n = 14,921) [1] | | | P value [1] | N | Population with CS used (n = 4794) [2] | | P value [2] |
|---|---|---|---|---|---|---|---|---|---|
| | | No. (%) | NO CS (n = 9,659) | WITH CS (n = 5,262) | | | Low-dose CS (n = 2,578) No (%) | CS Megadoses (n = 2,216) No (%) | |
| Median [range], Age (years) | 14,921 | 69.33 [56.34–79.9] | 67.5 [53.95–79.23] | 71.71 [60.14–80.97] | <0.001 | 4,794 | 73.79 [61.41–83.41] | 69.78 [59.29–79.10] | <0.001 |
| Age groups: | | | | | <0.001 | | | | <0.001 |
| < 40 years | | 927 (6.2) | 754 (7.8) | 173 (3.3) | | | 79 (3.1) | 71 (3.2) | |
| 40–50 years | | 1,433 (9.6) | 1,083 (11.2) | 350 (6.7) | | | 148 (5.7) | 165 (7.5) | |
| 50–60 years | | 2,385 (16.0) | 1,614 (16.7) | 771 (14.7) | | | 340 (13.2) | 354 (16.0) | |
| 60–70 years | | 2,923 (19.6) | 1,822 (18.9) | 1,101 (20.9) | | | 473 (18.4) | 534 (24.1) | |
| 70–80 years | | 3,570 (23.9) | 2,135 (22.1) | 1,435 (27.3) | | | 700 (27.2) | 603 (27.2) | |
| > 80 years | | 3,683 (24.7) | 2,251 (23.3) | 1,432 (27.2) | | | 838 (32.5) | 489 (22.1) | |
| Gender: | 14,906 | | | | <0.001 | 4,788 | | | <0.001 |
| Women | | 6,375 (42.8) | 4,443 (46.0) | 1,932 (36.8) | | | 1,044 (40.6) | 730 (33.0) | |
| Men | | 8,531 (57.2) | 5,208 (54.0) | 3,323 (63.2) | | | 1,530 (59.4) | 1,484 (67.0) | |
| Race: | 14,678 | | | | <0.001 | 4,708 | | | 0.43 |
| Caucasian | | 13,254 (90.3) | 8,527 (89.7) | 4,727 (91.4) | | | 2,316 (91.7) | 1,988 (91.1) | |
| African American | | 54 (0.4) | 33 (0.4) | 21 (0.4) | | | 12 (0.5) | 8 (0.4) | |
| Latin | | 1,182 (8.1) | 822 (8.7) | 360 (7.0) | | | 167 (6.6) | 162 (7.4) | |
| Asian | | 63 (0.4) | 47 (0.5) | 16 (0.3) | | | 5 (0.2) | 9 (0.4) | |
| Other | | 125 (0.9) | 79 (0.8) | 46 (0.9) | | | 25 (1.0) | 16 (0.7) | |
| Arterial hypertension | 14,899 | 7,573 (50.8) | 4,590 (47.6) | 2,983 (56.8) | <0.001 | 4,789 | 1,531 (59.5) | 1,215 (54.9) | 0.001 |
| Type 2 diabetes mellitus | 14,876 | 2,864 (19.3) | 1,724 (17.9) | 1,140 (21.7) | <0.001 | 4,791 | 573 (22.2) | 476 (21.5) | 0.54 |
| Dyslipidaemia | 14,890 | 5,902 (39.6) | 3,630 (37.7) | 2,272 (43.3) | <0.001 | 4,783 | 1,169 (45.5) | 908 (41.1) | 0.002 |
| Obesity (BMI>30) | 13,573 | 2,866 (21.1) | 1,657 (18.8) | 1,209 (25.4) | <0.001 | 4,344 | 583 (25.1) | 539 (26.7) | 0.23 |
| Smoking status: | 14,227 | | | | <0.001 | 4,548 | | | 0.29 |
| Never | | 9,859 (69.3) | 6,626 (71.7) | 3,233 (64.8) | | | 1,585 (64.3) | 1,376 (66.1) | |
| Formed | | 3,613 (25.4) | 2,128 (23.0) | 1,485 (29.8) | | | 757 (30.7) | 594 (28.5) | |
| Current | | 755 (5.3) | 486 (5.3) | 269 (5.4) | | | 123 (5.0) | 113 (5.4) | |
| Atrial fibrillation | 14,881 | 1,663 (11.2) | 1,040 (10.8) | 623 (11.9) | 0.044 | 4,778 | 349 (13.6) | 218 (9.9) | <0.001 |
| Myocardial infarction | 14,886 | 1,188 (8.0) | 703 (7.3) | 485 (9.2) | <0.001 | 4,787 | 259 (10.1) | 186 (8.4) | 0.049 |
| Hearth failure | 14,893 | 1,071 (7.2) | 637 (6.6) | 434 (8.3) | <0.001 | 4,787 | 259 (10.1) | 133 (6.0) | <0.001 |
| COPD | 14,893 | 1,021 (6.9) | 505 (5.2) | 516 (9.8) | <0.001 | 4,784 | 302 (11.7) | 168 (7.6) | <0.001 |
| Chronic bronchitis | 14,891 | 746 (5.0) | 416 (4.3) | 330 (6.3) | <0.001 | 4,787 | 199 (7.7) | 105 (4.8) | <0.001 |
| Asthma | 14,888 | 1,079 (7.3) | 652 (6.8) | 427 (8.1) | 0.002 | 4,789 | 235 (9.1) | 153 (6.9) | 0.005 |
| Obstructive Sleep Apnea Syndrome | 14,825 | 884 (6.0) | 494 (5.1) | 390 (7.5) | <0.001 | 4,768 | 184 (7.2) | 176 (8.0) | 0.29 |
| Dementia | 14,890 | 1,496 (10.1) | 994 (10.3) | 502 (9.6) | 0.14 | 4,788 | 335 (13.0) | 135 (6.1) | <0.001 |
| Stroke | 14,873 | 1,081 (7.3) | 664 (6.9) | 417 (7.9) | 0.019 | 4,782 | 235 (9.2) | 151 (6.8) | 0.003 |
| Neurodegenerative disease | 14,897 | 1,356 (9.1) | 869 (9.0) | 487 (9.3) | 0.59 | 4,784 | 316 (12.3) | 142 (6.4) | <0.001 |
| Cancer | 14,878 | 1,241 (8.3) | 787 (8.2) | 454 (8.6) | 0.32 | 4,787 | 234 (9.1) | 183 (8.3) | 0.32 |
| Leukaemia | 14,903 | 179 (1.2) | 93 (1.0) | 86 (1.6) | <0.001 | 4,791 | 43 (1.7) | 37 (1.7) | 0.99 |
| Lymphoma | 14,892 | 212 (1.4) | 120 (1.3) | 92 (1.8) | 0.013 | 4,789 | 45 (1.8) | 35 (1.6) | 0.65 |
| HIV infection | 14,861 | 102 (0.7) | 73 (0.8) | 29 (0.6) | 0.15 | 4,779 | 17 (0.7) | 11 (0.5) | 0.46 |

CS = Corticosteroids, BMI = Body Mass Index, COPD = Chronic Obstructive Pulmonar Disease, HIV = Human Inmunodefiency Virus.

[1]: Bivariate analysis with the total population, hypothesis testing according to the use or not of corticosteroids.

[2]: Bivariate analysis only with patients in whom corticosteroids were used and we have information on the use of megadoses, hypothesis testing according to the use or not of megadoses of corticosteroids.

**Table 2. Use of megadoses according to analytical parameters and radiological findings on admission.**

| | NO CS | | Low-dose CS | | CS Megadoses | | P Value | P Value [1] | | |
| | No. | Mean (SD) | No. | Mean (SD) | No. | Mean (SD) | | A vs B | A vs C | B vs C |
|---|---|---|---|---|---|---|---|---|---|---|
| Haemoglobin (g/dL) | 10,042 | 13.72 (1.86) | 2,566 | 13.50 (2.0) | 2,209 | 13.83 (1.92) | <0.001 | <0.001 | 0.007 | <0.001 |
| Platelets (x 10^6/L) | 10,047 | 207.428(92.308) | 2,563 | 201.961(90.659) | 2,203 | 210.757 (96.110) | 0.001 | 0.001 | 0.597 | 0.003 |
| Leukocytes (x 10^6/L) | 1,040 | 7137 (5283) | 2,566 | 7917(5807) | 2,209 | 7837 (6077) | <0.001 | <0.001 | <0.001 | 0.259 |
| Neutrophils (x 10^6/L) | 9,987 | 5262 (4627) | 2,558 | 6125 (4501) | 2,201 | 6030 (4472) | <0.001 | <0.001 | <0.001 | 0.622 |
| Lymphocytes (x 10^6/L) | 10,026 | 1182 (1981) | 2,561 | 1156 (2695) | 2,206 | 1117 (2362) | <0.001 | <0.001 | <0.001 | 0.754 |
| CPR (mg/L) | 9,699 | 76.65 (83.64) | 2,459 | 105 (93.55) | 2,137 | 114.2 (96.73) | <0.001 | <0.001 | <0.001 | <0.001 |
| Procalcitonin (ng/mL) | 4,625 | 0.4341(2.353) | 1,312 | 0.6354 (2.361) | 1,112 | 0.4706 (1.991) | <0.001 | <0.001 | <0.001 | 0.230 |
| Lactate dehydrogenase (U/L) | 8,654 | 355.7 (207.6) | 2,208 | 390.1 (214) | 2,023 | 419.9 (284.4) | <0.001 | <0.001 | <0.001 | <0.001 |
| Interleukin-6 (pg/mL) | 1,146 | 54.92 (160.4) | 3,57 | 82,57 (158,1) | 4,65 | 91.65 (218.9) | <0.001 | <0.001 | <0.001 | 0.386 |
| D-dimer (ng/mL) | 7,631 | 1643 (7960) | 2,034 | 2458 (12445) | 1,963 | 2417 (11879) | <0.001 | <0.001 | <0.001 | 0.267 |
| | N | No (%) | N | No (%) | N | No (%) | | | | |
| Condensation | 9,988 | 4,753 (47.6) | 2,544 | 1,298 (51.0) | 2,207 | 1,152 (52.2) | <0.001 | 0.002 | <0.001 | 0.432 |
| Interstitial infiltrate | 9,989 | 6,050 (60.6) | 2,552 | 1,617(63.4) | 2205 | 1570 (71.2) | <0.001 | 0.010 | <0.001 | <0.001 |
| Pleural effusion | 9,979 | 458 (4.6) | 2,553 | 137 (5.4) | 2,206 | 89 (4) | <0.001 | 0.106 | 0.280 | 0.034 |

CPR = C-reactive protein, CS = Costicosteroids, SD = Standard Desviation.

[1]: P-value of the hypothesis test for subgroups. Mann Whitney U or Fisher's exact test was used as appropriate.

Groups A: No CS; B: Low-dose CS; C: CS-Megadoses.

corticosteroid group it is also associated with higher mortality (OR 1.40, 95% CI 1.21–1.61; p < .001). There is no difference in mortality between megadoses and low-dose (p .298).

In addition, when analyzing the relationship with mortality, patients who received lopinavir-ritonavir (OR 0.65, 95% CI 0.58–0.74; p < .001), hydroxychloroquine (OR 0.53, 95% CI 0.44–0.63; p < .001) and tocilizumab (OR 0.84, 95% CI 0.71–0.98; p .029), among others, had a higher survival rate.

## Discussion

Throughout these long months of the covid19 pandemic, there has been much controversy about the role of corticosteroids in covid19 pneumonia, as there was no evidence of benefit in

**Table 3. Other immunomodulatory therapies used in patients with CS.**

| | WITH CS No. (Total n = 5,262) | Low-dose CS (n = 2,578) No (%) | CS Megadoses (n = 2,216) No (%) | P value |
|---|---|---|---|---|
| Use of lopinavir-ritonavir | 3,082 (4,784) | 1,499 (58.3) | 1,583 (71.5) | <**0.001** |
| Use of hidroxychloroquine | 4,284 (4,789) | 2,260 (87.7) | 2,024 (91.5) | <**0.001** |
| Use of beta-interferon 1B | 669 (4,768) | 376 (14.7) | 293 (13.3) | 0.17 |
| Use of tocilizumab | 842 (4,775) | 325 (12.7) | 517 (23.5) | <**0.001** |
| Use of anakinra | 82 (4,751) | 17 (0.7) | 65 (3.0) | <**0.001** |
| Use of remdesivir | 38 (4,755) | 24 (0.9) | 14 (0.6) | 0.25 |
| Use of chloroquine | 213 (4,765) | 79 (3.1) | 134 (6.1) | <**0.001** |
| Use of immunoglobulins | 53 (4,725) | 12 (0.5) | 41 (1.9) | <**0.001** |
| Use of baricitinib | 86 (3,929) | 15 (0.7) | 71 (3.8) | <**0.001** |
| Use of colchicine | 67 (4,732) | 40 (1.6) | 27 (1.2) | 0.31 |
| Use of inhaled beclomethasone | 355 (4,746) | 191 (7.5) | 164 (7.5) | 0.98 |

CS = Corticosteroids.

**Table 4. Development of complications in patients with CS.**

| | WITH CS No. (Total n = 5,262) | Low-dose CS (n = 2,578) No (%) | CS Megadoses (n = 2,216) No (%) | Odds ratio (IC 95%) | P value |
|---|---|---|---|---|---|
| Heart failure | 4,789 | 256 (9.9) | 132 (6.0) | 0.57 (0.46–0.72) | **<0.001** |
| Cardiac arrhytmia | 4,787 | 171 (6.6) | 113 (5.1) | 0.76 (0.59–0.97) | **0.026** |
| Epileptic seizure | 4,790 | 23 (0.9) | 13 (0.6) | 0.66 (0.33–1.30) | 0.23 |
| Stroke | 4,787 | 14 (0.5) | 31 (1.4) | 2.60 (1.38–4.90) | **0.003** |
| Acute renal failure | 4,786 | 542 (21.1) | 401 (18.1) | 0.83 (0.72–0.96) | **0.010** |
| Venous thromboembolic disease | 4,781 | 72 (2.8) | 104 (4.7) | 1.72 (1.26–2.33) | **0.001** |
| Acute peripheral arterial disease | 4,765 | 17 (0.7) | 16 (0.7) | 1.09 (0.55–2.17) | 0.80 |
| Disseminated intravascular coagulation | 4,782 | 40 (1.6) | 43 (2.0) | 1.26 (0.81–1.94) | 0.30 |
| Bacterial pneumonia | 4,784 | 487 (18.9) | 356 (16.1) | 0.82 (0.71–0.95) | **0.010** |
| Sepsis | 4,785 | 260 (10.1) | 182 (8.2) | 0.80 (0.65–0.97) | **0.026** |
| Shock | 4,776 | 203 (7.9) | 142 (6.4) | 0.80 (0.65–1.01) | 0.051 |
| Multiorgan failure | 4,782 | 223 (8.7) | 180 (8.2) | 0.93 (0.76–1.15) | 0.52 |

CS = Corticosteroids.

previous similar viral infections [5]. In March 2020, the World Health Organization (WHO) advised against its use [7], however the results of the RECOVERY clinical trial showed a reduction in mortality in the patients treated with 10 days of dexamethasone 6 mg compared to the placebo group [8].

To begin with, it is worth highlighting in this study the high number of patients who received corticosteroids. Specifically, 2216 out of 4794 patients received megadoses, considering megadose an amount of prednisone greater than 150 mg per day [9], representing a 46% in comparison with the 40% of people receiving pulse therapy in the Irastorza et al. observational

**Table 5. Outcome according to the use of megadoses.**

| | WITH CS No. (Total n = 5,262) | Low-dose CS (n = 2,578) No (%) | CS Megadoses (n = 2,216) No (%) | Odds ratio (IC 95%) | P value |
|---|---|---|---|---|---|
| Hospital stay in days, median (IQR) | 4,794 | 12 (7–18) | 12 (8–19) | 0.99 (0.99–1.01) | 0.88 |
| High-flow nasal cannula | 4,767 | 325 (12.7) | 340 (15.4) | 1.25 (1.06–1.47) | **0.008** |
| Non-invasive mechanical ventilation | 4,778 | 214 (8.3) | 243 (11.0) | 1.36 (1.12–1.65) | **0.002** |
| Invasive mechanical ventilation | 4,782 | 307 (12.0) | 250 (11.3) | 0.94 (0.79–1.12) | 0.49 |
| Prone position | 4,774 | 413 (16.1) | 586 (26.6) | 1.89 (1.64–2.17) | **<0.001** |
| ICU admission | 4,793 | 370 (14.4) | 332 (15.0) | 1.05 (0.90–1.23) | 0.54 |
| *Resolution of first episode* | | | | | |
| Discharge home | 4,794 | 1,656 (64.2) | 1,509 (68.1) | 1 (ref.) | - |
| Convalescence centre | | 139 (5.4) | 108 (4.9) | 0.85 (0.66–1.11) | 0.23 |
| Death during hospital admission | | 783 (30.4) | 599 (27.0) | 0.84 (0.74–0.95) | **0.007** |
| *Readmission to hospital* | | | | | |
| Readmission | 4,630 | 130 (5.2) | 75 (3.5) | 0.66 (0.49–0.88) | **0.005** |
| Days of discharge to readmission, median (IQR) | 205 | 8.5 (2–16) | 12 (5–15) | 1.02 (0.99–1.05) | 0.12 |
| Mortality [1] | 4,750 | 803 (31.4) | 608 (27.7) | 0.84 (0.74–0.95) | **0.005** |

ICU = Intensive care unit, CS = Corticosteroids.

[1]: Death at any time. Either on first admission, on discharge or on re-admission.

**Table 6. Corticosteroids therapy and mortality.**

| | No. (Total n = 14,921) | No. (%) | SURVIVORS (n = 11,862) | DECEASED (n = 3,059) | Odds ratio (IC 95%) | P value |
|---|---|---|---|---|---|---|
| Use of systemic corticosteroids | 5,262 | 35.3 | 3,763 (31.7) | 1,499 (49) | 2.07 (1.91–2.24) | **<0.001** |
| Use of CS Megadoses | 2,216 | 46.2 | 1617 (47.4) | 599 (43.3) | 0.84 (0.75–0.96) | **0.011** |
| **Days from sympton onset to start of corticosteroids** | | | | | | |
| < 10 days | 5,023 | 2,719 (54.1) | 1,783 (49.5) | 936 (65.9) | 1 (ref.) | - |
| > 10 days | | 2,304 (45.9) | 1,820 (50.5) | 484 (34.1) | 0.51 (0.45–0.58) | **<0.001** |
| | OTHER IMMUNOMODULATORY THERAPIES USED IN PATIENTS WITH CS | | | | | |
| Use of lopinavir-ritonavir | 9,148 | 61.4 | 1599 (52.4) | 7549 (63.7) | 0.63 (0.58–0.68) | **<0.001** |
| Use of hidroxychloroquine | 12,772 | 85.7 | 10487 (88.5) | 2285 (74.7) | 0.34 (0.35–0.43) | **<0.001** |
| Use of tocilizumab | 1,257 | 8.4 | 948 (8) | 309 (10.1) | 1.294 (1.13–1.48) | **<0.001** |
| Use of baricitinib | 92 | 0.8 | 80 (0.9) | 12 (0.5) | 0.5707 (0.31–1.04) | **0.071** |
| | OTHER TREATMENT STRATEGIES EMPLOYED | | | | | |
| High-flow nasal cannula | 1,189 | 8 | 767 (6.5) | 422 (13.9) | 2.32 (2.04–2.63) | **<0.001** |
| Non-invasive mechanical ventilation | 719 | 4.8 | 351 (3) | 368 (12.1) | 4.5 (3.86–5.23) | **<0.001** |
| Invasive mechanical ventilation | 975 | 6.6 | 535 (4.5) | 440 (14.4) | 3.56 (3.12–4.07) | **<0.001** |
| Prone position | 1,519 | 10.2 | 854 (7.2) | 665 (21.9) | 3.6 (3.2–4.01) | **<0.001** |
| ICU admission | 1,218 | 8.2 | 737 (6.2) | 481 (15.7) | 2.8 (2.5–3.2) | **<0.001** |

ICU = Intensive care unit, CS = Corticosteroids.

**Table 7. Megadoses and mortality (multivariate analysis adjusted according to patient age, comorbidities and other treatments).**

| | Odds ratio (IC 95%) | P value |
|---|---|---|
| **Corticosteroids therapy** | | |
| No CS | 1 (ref.) | - |
| Low-dose CS | 1.40 (1.21–1.61) | **<0.001** |
| CS Megadoses | 1.54 (1.32–1.80) | **<0.001** |
| Age | 1.09 (1.09–1.10) | **<0.001** |
| Sex (women) | 0.59 (0.52–0.66) | **<0.001** |
| Arterial hypertension | 1.21 (1.07–1.37) | **0.002** |
| Dyslipidaemia | 1.08 (0.97–1.21) | **0.145** |
| Atrial fibrillation | 1.2 (1.03–1.40) | **0.017** |
| Hearth failure | 1.47 (1.22–1.77) | **<0.001** |
| COPD | 1.28 (1.07–1.54) | **0.07** |
| Stroke | 1.25 (1.05–1.48) | **0.012** |
| Dementia | 1.30 (1.05–1.61) | **0.015** |
| Neurodegenerative disease | 1.33 (1.07–1.66) | **0.010** |
| Use of lopinavir-ritonavir | 0.99 (0.88–1.12) | **0.93** |
| Use of hydroxychloroquine | 0.50 (0.43–0.57) | **<0.001** |
| Use of tocilizumab | 0.62 (0.49–0.80) | **<0.001** |
| Use of baricitinib | 0.34 (0.16–0.71) | **0.005** |
| High-flow nasal cannula | 1.73 (1.41–2.11) | **<0.001** |
| Non-invasive mechanical ventilation | 4.01 (3.17–5.07) | **<0.001** |
| Invasive mechanical ventilation | 5.32 (3.16–8.98) | **<0.001** |
| Prone position | 3.33 (2.71–4.08) | **<0.001** |
| ICU admission | 0.68 (0.41–1.45) | **0.151** |

ICU = Intensive care unit, CS = Corticosteroids, COPD = Chronic Obstructive Pulmonar Disease.

study of 242 patients [10], and the 20% described in the one published by López Zuñiga with 318 participants [11].

The particular interest in evaluating the difference between megadoses in contrast to lower doses was based on the hypothesis that they could have different impact on the evolution of the disease, as well as different side effects, since pulse therapy uses the non-genomic mechanisms of glucocorticoids to enhance the anti-inflammatory power and reduce the metabolic side effects and incidence of infections [12], as it has been demonstrated in other systemic autoimmune diseases [13].

Even though the advanced age, hypertension and dyslipidemia were postulated as risk factors for ADRS [14], in our work the patients receiving megadoses were younger and suffered less comorbidities than the patients of the other group, in contrast to the Irastorza series [10] in which no significant differences were observed. However, in his study, patients who received pulse therapy out of the second week of the disease and patients who did not receive corticosteroids were included in the same group. Returning to our study, a significant greater number of men received megadoses compared to women, 67% vs 33% respectively, perhaps because of the serious incidence of ADRS in men and their worse prognosis compared to women [15].

It should also be noticed that patients receiving previous immunosuppressive therapy received lower proportion of megadoses; we could not know if this was due to the fear of viral persistence already described in immunocompromised hosts [16], or because of a milder course in these patients, as current evidence only shows a probably increased risk of severe COVID-19 and death in patients with malignancy or solid organ transplant recipients, but this is less clear in other immunocompromised patients [17].

As for the situation at admission, the patients receiving megadoses had significant higher levels of lactate-dehydrogenase, c-reactive-protein and D-Dimer in contrast to the low-dose group, considering the higher these inflammatory markers are, the more they have been associated with lung damage, ADRS and worse prognosis. We found similar results about the radiological scenario, since the people treated with megadoses had a significant interstitial infiltrate in the X ray at admission in comparison with the low-dose group, and the radiological extension has also been associated to ADRS [18].

Regarding concomitant use with other immunomodulatory treatments or adjunctive treatments such us lopinavir-ritonavir, the group of patients of megadoses treatment received also more lopinavir-ritonavir, tocilizumab and bariticinib. A sub-analysis in our study of their effect on mortality showed that patients on lopinavir-ritonavir survived longer, as did those on tocilizumab and bariticinib. In this sense, it was hypothesized whether the concomitant effect of both, megadoses of corticosteroids and the specific immunomodulator, may influence the survival of COVID-19 patients. On tocilizumab, the EMPACTA clinical trial [19] included 249 patients in the tocilizumab group and 128 patients in the placebo group and the results suggested that patients who were most likely to benefit from tocilizumab had moderate or severe disease and that tocilizumab may add a potential benefit to antiviral treatment and glucocorticoids. Concerning bariticinib, there are few studies reflecting its use and impact on covid-19 and they include few patients. [20,21] In our study there were 86 cases registered who received corticosteroids at the same time, mostly megadoses, with a protective effect on mortality which is an interesting finding that requires further study. As for lopinavir-ritonavir, a randomized trial found that this treatment added to standard supportive care was not associated with clinical improvement or mortality in seriously ill patients with COVID-19 compared to standard care alone [22].

As for the development of complications during admission, significantly more complications of heart failure, arrhythmias and renal failure were observed in the non-megadose group,

probably influenced by the higher proportion of comorbidities observed in this group compared to the megadose group. On the other hand, the incidence of venous thromboembolic disease and stroke in patients who used megadoses was higher, maybe explained by the greater inflammation in these patients, as it has been demonstrated in other studies [23,24]. In other series, there have been reported a 7% of bacterial coinfections in hospitalized COVID-19 patients, increasing to 14% in studies that only included ICU patients [25]. We would like to highlight that no higher proportion of bacterial pneumonia or sepsis were observed in the megadose group, supporting the initial hypothesis on the use of the non-genomic pathway of megadoses, as explained earlier [12].

Regarding the evolution of the patients who received corticosteroids, no increase in the average hospital stay was described among those who used megadoses. The patients in the megadoses group required more high-flow devices and non-invasive mechanical ventilation, but there were no differences between groups in terms of transfers to ICU or invasive ventilation.

In the observational study of Fernandez Cruz et al, a significant reduction in mortality was demonstrated among glucocorticoids users in the group classified as moderate-severe disease, but there were not significant differences between the patients receiving 1 mg/kg/d of methylprednisolone or pulse therapy (up to 500mg/d) [26]. In this study, the preliminary bivariate analysis showed an increased mortality among the patients receiving corticosteroids, however, in the group treated with megadoses (OR 0.85 CI 0.75–0.96) the survival rate was higher compared to the no megadoses group. The statistical significance disappeared in the multivariate analysis due to the introduction of confounding factors. Increased mortality is observed in the megadose group, as opposed to this type of regimen in other systemic autoimmune diseases [13]. Several studies have been recently published showing the effectiveness of high dose glucocorticoid pulse therapy in the prognosis of patients with COVID19 pneumonia in the inflammatory stage of the disease [10,11,27] in contrast to a Brazilian double-blind, randomized, placebo-controlled trial which reported no benefit from the use of methylprednisolone [28].

Thus, there is still no clear answer to which dose should be used, how long it should last or even if there are significant differences between dexamethasone and methylprednisolone. A new randomized controlled trial (CORTIVID) is coming soon with the intention to evaluate the role of pulse therapy [29].

## Conclusion

This study includes a huge number of patients treated with corticosteroids and specifically with megadoses. There is no difference in mortality with megadoses versus low-dose of corticosteroids, but there is a lower incidence of infectious complications in megadoses group.

## Limitations

It is a retrospective study. We could not evaluate the impact of megadoses in the respiratory situation, radiological evolution, nor in the inflammatory parameters, as we only had the data at the moment of hospital admission and a week later, so we could not establish a direct relationship with the glucocorticoid treatment, since it is difficult to establish the temporal relationship between the evolution of the evolution of the clinical parameters and the treatment. Besides, although the treatment regimens were divided into megadoses vs low-dose of corticosteroids based on > 150mg of prednisone/day or <150mg/day respectively, we could not establish the exact treatment regimens, the type of glucocorticoid used, nor its duration. Moreover, there are other glucocorticoid-related infections and side effects that have not been evaluated in this registry, so further studies and specific clinical trials to evaluate the differences between regimens are needed.

## Supporting information

**S1 File. Statistical results.**
(DOCX)

**S2 File.**
(DOCX)

## Acknowledgments

We gratefully acknowledge all the investigators who participate in the SEMI-COVID-19 Registry. The authors declare that there are no conflicts of interest.

List of the SEMI-COVID-19 Network members:

Coordinator of the SEMI-COVID-19 Registry: José Manuel Casas Rojo. jm.casas@gmail.com

SEMI-COVID-19 Scientific Committee Members: José Manuel Casas Rojo, José Manuel Ramos Rincón, Carlos Lumbreras Bermejo, Jesús Millán Núñez-Cortés, Juan Miguel Antón Santos, Ricardo Gómez Huelgas.

Members of the SEMI-COVID-19 Group

H. Univ. de Bellvitge. L'Hospitalet de Llobregat (Barcelona):

Xavier Corbella, Narcís Homs, Abelardo Montero, Jose María Mora-Luján, Manuel Rubio-Rivas

H. U. 12 de Octubre. Madrid:

Paloma Agudo de Blas, Coral Arévalo Cañas, Blanca Ayuso, José Bascuñana Morejón, Samara Campos Escudero, María Carnevali Frías, Santiago Cossio Tejido, Borja de Miguel Campo, Carmen Díaz Pedroche, Raquel Diaz Simon, Ana García Reyne, Laura Ibarra Veganzones, Lucia Jorge Huerta, Antonio Lalueza Blanco, Jaime Laureiro Gonzalo, Jaime Lora-Tamayo, Carlos Lumbreras Bermejo, Guillermo Maestro de la Calle, Rodrigo Miranda Godoy, Barbara Otero Perpiña, Diana Paredes Ruiz, Marcos Sánchez Fernández, Javier Tejada Montes

H. U. Gregorio Marañon. Madrid:

Laura Abarca Casas, Álvaro Alejandre de Oña, Rubén Alonso Beato, Leyre Alonso Gonzalo, Jaime Alonso Muñoz, Crhistian Mario Amodeo Oblitas, Cristina Ausín García, Marta Bacete Cebrián, Jesús Baltasar Corral, Maria Barrientos Guerrero, Alejandro D. Bendala Estrada, María Calderón Moreno, Paula Carrascosa Fernández, Raquel Carrillo, Sabela Castañeda Pérez, Eva Cervilla Muñoz, Agustín Diego Chacón Moreno, Maria Carmen Cuenca Carvajal, Sergio de Santos, Andrés Enríquez Gómez, Eduardo Fernández Carracedo, María Mercedes Ferreiro-Mazón Jenaro, Francisco Galeano Valle, Alejandra Garcia, Irene Garcia Fernandez-Bravo, María Eugenia García Leoni, María Gómez Antúnez, Candela González San Narciso, Anthony Alexander Gurjian, Lorena Jiménez Ibáñez, Cristina Lavilla Olleros, Cristina Llamazares Mendo, Sara Luis García, Víctor Mato Jimeno, Clara Millán Nohales, Jesús Millán Núñez-Cortés, Sergio Moragón Ledesma, Antonio Muiño Míguez, Cecilia Muñoz Delgado, Lucía Ordieres Ortega, Susana Pardo Sánchez, Alejandro Parra Virto, María Teresa Pérez Sanz, Blanca Pinilla Llorente, Sandra Piqueras Ruiz, Guillermo Soria Fernández-Llamazares, María Toledano Macías, Neera Toledo Samaniego, Ana Torres do Rego, Maria Victoria Villalba Garcia, Gracia Villarreal, María Zurita Etayo

H. Costa del Sol. Marbella (Málaga):

Victoria Augustín Bandera, Javier García Alegría, Nicolás Jiménez-García, Jairo Luque del Pino, María Dolores Martín Escalante, Francisco Navarro Romero, Victoria Nuñez Rodriguez, Julián Olalla Sierra

H. de Cabueñes. Gijón (Asturias):

Ana María Álvarez Suárez, Carlos Delgado Vergés, Rosa Fernandez-Madera Martínez, Eva Mª Fonseca Aizpuru, Alejandro Gómez Carrasco, Cristina Helguera Amezua, Juan Francisco

López Caleya, Diego López Martínez, María del Mar Martínez López, Aleida Martínez Zapico, Carmen Olabuenaga Iscar, Lucía Pérez Casado, María Luisa Taboada Martínez, Lara María Tamargo Chamorro

H. U. La Paz. Madrid:

Jorge Álvarez Troncoso, Francisco Arnalich Fernández, Francisco Blanco Quintana, Carmen Busca Arenzana, Sergio Carrasco Molina, Aranzazu Castellano Candalija, Germán Daroca Bengoa, Alejandro de Gea Grela, Alicia de Lorenzo Hernández, Alejandro Díez Vidal, Carmen Fernández Capitán, Maria Francisca García Iglesias, Borja González Muñoz, Carmen Rosario Herrero Gil, Juan María Herrero Martínez, Víctor Hontañón, Maria Jesús Jaras Hernández, Carlos Lahoz, Cristina Marcelo Calvo, Juan Carlos Martín Gutiérrez, Monica Martinez Prieto, Elena Martínez Robles, Araceli Menéndez Saldaña, Alberto Moreno Fernández, Jose Maria Mostaza Prieto, Ana Noblejas Mozo, Carlos Manuel Oñoro López, Esmeralda Palmier Peláez, Marina Palomar Pampyn, Maria Angustias Quesada Simón, Juan Carlos Ramos Ramos, Luis Ramos Ruperto, Aquilino Sánchez Purificación, Teresa Sancho Bueso, Raquel Sorriguieta Torre, Clara Itziar Soto Abanedes, Yeray Untoria Tabares, Marta Varas Mayoral, Julia Vásquez Manau

H. Royo Villanova. Zaragoza:

Nicolás Alcalá Rivera, Anxela Crestelo Vieitez, Esther del Corral Beamonte, Jesús Díez Manglano, Isabel Fiteni Mera, Maria del Mar Garcia Andreu, Martin Gericó Aseguinolaza, Cristina Gallego Lezaun, Claudia Josa Laorden, Raul Martínez Murgui, Marta Teresa Matía Sanz

H. Reg. Univ. de Málaga:

Mª Mar Ayala-Gutiérrez, Rosa Bernal López, José Bueno Fonseca, Verónica Andrea Buonaiuto, Luis Francisco Caballero Martínez, Lidia Cobos Palacios, Clara Costo Muriel, Francis de Windt, Ana Teresa Fernandez-Truchaud Christophel, Paula García Ocaña, Ricardo Gómez Huelgas, Javier Gorospe García, José Antonio Hurtado Oliver, Sergio Jansen-Chaparro, Maria Dolores López-Carmona, Pablo López Quirantes, Almudena López Sampalo, Elizabeth Lorenzo-Hernández, Juan José Mancebo Sevilla, Jesica Martín Carmona, Luis Miguel Pérez-Belmonte, Iván Pérez de Pedro, Araceli Pineda-Cantero, Carlos Romero Gómez, Michele Ricci, Jaime Sanz Cánovas

H. Clínico de Santiago de Compostela (A Coruña):

Maria del Carmen Beceiro Abad, Maria Aurora Freire Romero, Sonia Molinos Castro, Emilio Manuel Paez Guillan, María Pazo Nuñez, Paula Maria Pesqueira Fontan

H. Universitario Dr. Peset. Valencia:

Juan Alberto Aguilera Ayllón, Arturo Artero, María del Mar Carmona Martín, María José Fabiá Valls, Maria de Mar Fernández Garcés, Ana Belén Gómez Belda, Ian López Cruz, Manuel Madrazo López, Elisabeth Mateo Sanchis, Jaume Micó Gandia, Laura Piles Roger, Adela Maria Pina Belmonte, Alba Viana García

H. Moisès Broggi. Sant Joan Despí (Barcelona):

Judit Aranda Lobo, Lucía Feria Casanovas, Jose Loureiro Amigo, Miguel Martín Fernández, Isabel Oriol Bermúdez, Melani Pestaña Fernández, Nicolas Rhyman, Nuria Vázquez Piqueras

C. H. U. de Badajoz:

Rafael Aragon Lara, Inmaculada Cimadevilla Fernandez, Juan Carlos Cira García, Gema Maria García García, Julia Gonzalez Granados, Beatriz Guerrero Sánchez, Francisco Javier Monreal Periáñez, Maria Josefa Pascual Perez

H. U. Río Hortega. Valladolid:

Irene Arroyo Jiménez, Marina Cazorla González, Marta Cobos-Siles, Luis Corral-Gudino, Pablo Cubero-Morais, María González Fernández, José Pablo Miramontes González, Marina Prieto Dehesa, Pablo Sanz Espinosa

H. U. Reina Sofía. Córdoba:

Antonio Pablo Arenas de Larriva, Pilar Calero Espinal, Javier Delgado Lista, Francisco Fuentes-Jiménez, María del Carmen Guerrero Martínez, María Jesús Gómez Vázquez, Jose Jiménez Torres, Laura Limia Pérez, José López-Miranda, Laura Martín Piedra, Marta Millán Orge, Javier Pascual Vinagre, Pablo Pérez-Martinez, María Elena Revelles Vílchez, Angela Rodrigo Martínez, Juan Luis Romero Cabrera, José David Torres-Peña.

H. Nuestra Señora del Prado. Talavera de la Reina (Toledo):

Sonia Casallo Blanco, Jeffrey Oskar Magallanes Gamboa, Cristina Salazar Mosteiro, Andrea Silva Asiain

H. U. S. Juan de Alicante (Alicante):

Marisa Asensio Tomás, David Balaz, David Bonet Tur, Ruth Cañizares Navarro, Paloma Chazarra Pérez, Jesús Corbacho Redondo, Eliana Damonte White, María Escamilla Espínola, Leticia Espinosa Del Barrio, Pedro Jesús Esteve Atiénzar, Carles García Cervera, David Francisco García Núñez, Francisco Garrido Navarro, Vicente Giner Galvañ, Angie Gómez Uranga, Javier Guzmán Martínez, Isidro Hernández Isasi, Lourdes Lajara Villar, Verónica Martínez Sempere, Juan Manuel Núñez Cruz, Sergio Palacios Fernández, Juan Jorge Peris García, Rafael Piñol Pleguezuelos, Andrea Riaño Pérez, José Miguel Seguí Ripoll, Azucena Sempere Mira, Philip Wikman-Jorgensen

H. G. U. de Elda (Alicante):

Carmen Cortés Saavedra, Jennifer Fernández Gómez, Borja González López, María Soledad Hernández Garrido, Ana Isabel López Amorós, Santiago López Gil, Maria de los Reyes Pascual Pérez, Nuria Ramírez Perea, Andrea Torregrosa García

H. U. Infanta Cristina. Parla (Madrid):

Juan Miguel Antón Santos, Ana Belén Barbero Barrera, Blanca Beamonte Vela, Coralia Bueno Muiño, Charo Burón Fernández, Ruth Calderón Hernáiz, Irene Casado López, José Manuel Casas Rojo, Andrés Cortés Troncoso, Pilar Cubo Romano, Francesco Deodati, Alejandro Estrada Santiago, Gonzalo García Casasola Sánchez, Elena García Guijarro, Francisco Javier García Sánchez, Pilar García de la Torre, Mayte de Guzmán García-Monge, Davide Luordo, María Mateos González, José A. Melero Bermejo, Cruz Pastor Valverde, José Luis Pérez Quero, Fernando Roque Rojas, Lorea Roteta García, Elena Sierra Gonzalo, Francisco Javier Teigell Muñoz, Juan Vicente de la Sota, Javier Villanueva Martínez

H. Santa Marina. Bilbao:

María Areses Manrique, Ainara Coduras Erdozain, Ane Labirua-Iturburu Ruiz

H. de Pozoblanco (Córdoba):

José Nicolás Alcalá Pedrajas, Antonia Márquez García, Inés Vargas

H. San Pedro. Logroño (La Rioja):

Diana Alegre González, Irene Ariño Pérez de Zabalza, Sergio Arnedo Hernández, Jorge Collado Sáenz, Beatriz Dendariena, Marta Gómez del Mazo, Iratxe Martínez de Narvajas Urra, Sara Martínez Hernández, Estela Menendez Fernández, Jose Luís Peña Somovilla, Elisa Rabadán Pejenaute

H. U. Son Llàtzer. Palma de Mallorca:

Andrés de la Peña Fernández, Almudena Hernández Milián

C. H. U. Ourense:

Raquel Fernández González, Amara Gonzalez Noya, Carlos Hernández Ceron, Isabel Izuzquiza Avanzini, Ana Latorre Diez, Pablo López Mato, Ana María Lorenzo Vizcaya, Daniel Peña Benítez, Milagros María Peña Zemsch, Lucía Pérez Expósito, Marta Pose Bar, Lara Rey González, Laura Rodrigo Lara

H. U. La Fe. Valencia:

Dafne Cabañero, María Calabuig Ballester, Pascual Císcar Fernández, Ricardo Gil Sánchez, Marta Jiménez Escrig, Cristina Marín Amela, Laura Parra Gómez, Carlos Puig Navarro, José Antonio Todolí Parra

H. de Mataró. Barcelona:

Raquel Aranega González, Ramon Boixeda, Javier Fernández Fernández, Carlos Lopera Mármol, Marta Parra Navarro, Ainhoa Rex Guzmán, Aleix Serrallonga Fustier

H. de Sagunto (Valencia):

Enrique Rodilla Sala, Jose María Pascual Izuel, Zineb Karroud Zamrani

H. Alto Guadalquivir. Andújar (Jaén):

Begoña Cortés Rodríguez

H. Infanta Margarita. Cabra (Córdoba):

María Esther Guisado Espartero, Lorena Montero Rivas, Maria de la Sierra Navas Alcántara, Raimundo Tirado-Miranda

C. H. U. de Ferrol (A Coruña):

Hortensia Alvarez Diaz, Tamara Dalama Lopez, Estefania Martul Pego, Carmen Mella Pérez, Ana Pazos Ferro, Sabela Sánchez Trigo, Dolores Suarez Sambade, Maria Trigas Ferrin, Maria del Carmen Vázquez Friol, Laura Vilariño Maneiro

H. U. Virgen del Rocío. Sevilla:

Reyes Aparicio Santos, Máximo Bernabeu-Wittel, Santiago Rodríguez Suárez, María Nieto, Luis Giménez Miranda, Rosa María Gámez Mancera, Fátima Espinosa Torre, Carlos Hernandez Quiles, Concepción Conde Guzmán, Juan Delgado de la Cuesta, Jara Eloisa Ternero Vega, María del Carmen López Ríos, Pablo Díaz Jiménez, Bosco Baron Franco, Carlos Jiménez de Juan, Sonia Gutiérrez Rivero, Julia Lanseros Tenllado, Verónica Alfaro Lara, Aurora González Estrada

H. Público de Monforte de Lemos (Lugo):

José López Castro, Manuel Lorenzo López Reboiro, Cristina Sardiña González

H. General Defensa. Zaragoza:

Anyuli Gracia Gutiérrez, Leticia Esther Royo Trallero

C. A. U. de Salamanca:

Gloria María Alonso Claudio, Víctor Barreales Rodríguez, Cristina Carbonell Muñoz, Adela Carpio Pérez, María Victoria Coral Orbes, Daniel Encinas Sánchez, Sandra Inés Revuelta, Miguel Marcos Martín, José Ignacio Martín González, José Ángel Martín Oterino, Leticia Moralejo Alonso, Sonia Peña Balbuena, María Luisa Pérez García, Ana Ramon Prados, Beatriz Rodríguez-Alonso, Ángela Romero Alegría, Maria Sanchez Ledesma, Rosa Juana Tejera Pérez

H. de Palamós (Girona):

Ana Alberich Conesa, Mari Cruz Almendros Rivas, Miquel Hortos Alsina, José Marchena Romero, Anabel Martin-Urda Diez-Canseco

H. Parc Tauli. Sabadell (Barcelona):

Francisco Epelde, Isabel Torrente

H. do Salnes. Vilagarcía de Arousa (Pontevedra):

Vanesa Alende Castro, Ana María Baz Lomba, Ruth Brea Aparicio, Marta Fernández Morales, Jesús Manuel Fernández Villar, María Teresa López Monteagudo, Cristina Pérez García, Lorena Rodríguez Ferreira, Diana Sande Llovo, Maria Begoña Valle Feijoo

H. U. HM Montepríncipe:

José F. Varona Arche

## Disclosure

**Consent for publication**. Only patients who had previously given consent for their medical records to be used for medical research were included in this registry. Data confidentiality and

patient anonymity were maintained at all times, in accordance with Spanish regulations on observational studies.

## Author Contributions

**Conceptualization:** Cristina Lavilla Olleros, Cristina Ausín García, Alejandro David Bendala Estrada, Ana Fernández Cruz, Jesús Millán Núñez-Cortés.

**Data curation:** Cristina Lavilla Olleros, Cristina Ausín García, Alejandro David Bendala Estrada, Jesús Millán Núñez-Cortés.

**Formal analysis:** Cristina Lavilla Olleros, Cristina Ausín García, Alejandro David Bendala Estrada, Jesús Millán Núñez-Cortés.

**Funding acquisition:** Cristina Lavilla Olleros, Cristina Ausín García, Alejandro David Bendala Estrada, Jesús Millán Núñez-Cortés.

**Investigation:** Cristina Lavilla Olleros, Cristina Ausín García, Alejandro David Bendala Estrada, Jesús Millán Núñez-Cortés.

**Methodology:** Cristina Lavilla Olleros, Cristina Ausín García, Alejandro David Bendala Estrada, Jesús Millán Núñez-Cortés.

**Project administration:** Cristina Lavilla Olleros, Cristina Ausín García, Alejandro David Bendala Estrada, José-Manuel Casas-Rojo, Jesús Millán Núñez-Cortés.

**Resources:** Cristina Lavilla Olleros, Cristina Ausín García, Alejandro David Bendala Estrada, Jesús Millán Núñez-Cortés.

**Software:** Cristina Lavilla Olleros, Cristina Ausín García, Alejandro David Bendala Estrada, Jesús Millán Núñez-Cortés.

**Supervision:** Cristina Lavilla Olleros, Cristina Ausín García, Alejandro David Bendala Estrada, Ana Muñoz, Ana Fernández Cruz, José-Manuel Casas-Rojo, Jesús Millán Núñez-Cortés.

**Validation:** Cristina Lavilla Olleros, Cristina Ausín García, Alejandro David Bendala Estrada, Philip Erick Wikman Jogersen, Ana Fernández Cruz, Vicente Giner Galvañ, Juan Antonio Vargas, José Miguel Seguí Ripoll, Manuel Rubio-Rivas, Rodrigo Miranda Godoy, Luis Mérida Rodrigo, Eva Fonseca Aizpuru, Francisco Arnalich Fernández, Arturo Artero, Jose Loureiro Amigo, Gema María García García, Luis Corral Gudino, Jose Jiménez Torres, José-Manuel Casas-Rojo, Jesús Millán Núñez-Cortés.

**Visualization:** Cristina Lavilla Olleros, Cristina Ausín García, Alejandro David Bendala Estrada, Philip Erick Wikman Jogersen, Ana Fernández Cruz, Jesús Millán Núñez-Cortés.

**Writing – original draft:** Cristina Lavilla Olleros, Cristina Ausín García, Alejandro David Bendala Estrada, Jesús Millán Núñez-Cortés.

**Writing – review & editing:** Cristina Lavilla Olleros, Cristina Ausín García, Alejandro David Bendala Estrada, Jesús Millán Núñez-Cortés.

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
