## [Decision Letter · Decision Letter 0]

21 Jul 2021

PONE-D-21-21800

Use of Glucocorticoids megadoses in SARS-CoV-2 infection in a spanish registry: SEMI-COVID-19.

PLOS ONE

Dear Dr. Lavilla Olleros,

Thank you for submitting your manuscript to PLOS ONE. After careful consideration, we feel that it has merit but does not fully meet PLOS ONE’s publication criteria as it currently stands. Therefore, we invite you to submit a revised version of the manuscript that addresses the points raised during the review process.

We look forward to receiving your revised manuscript.

Kind regards,

Aleksandar R. Zivkovic

Academic Editor

PLOS ONE

3. One of the noted authors is a group or consortium [SEMI-COVID-19 Network]. In addition to naming the author group, please list the individual authors and affiliations within this group in the acknowledgments section of your manuscript. Please also indicate clearly a lead author for this group along with a contact email address.

Reviewers' comments:

Reviewer #1: The authors present their experience with high-dose corticosteroids (CS), referred to as megadoses, and compare these against low-dose CS.

Intro, paragraph 2: consider rephrasing in order to be current with the literature. CS are the only intervention for severe COVID with consistent mortality impact across studies. I would avoid using the references 5-7 that raised controversies in CS use too early in the pandemic and based on theoretical assumptions from non-COVID diseases such as ARDS, SARS or MERS that have distinct etiology, epidemiology, phenotypes and outcomes. Instead, you can highlight even though CS have been widely adopted, there are still questions on dosing, timing and duration of CS that have not been systematically studied at large.

Methods: Study conclusion: consider renaming this section Study endpoints to avoid confusion with the use of the word conclusion. The Literature search section does not add to the methods, consider removing.

There are 2 definitions of megadoses, >150 mg of prednisone in 3-4 days and >150 mg of prednisone in 24h. Historically, high-dose CS in ARDS research has been defined as >200mg/Kg/day of methylprednisolone or its equivalent in other CS forms (prednisone, prednisolone, etc). If the authors choose to study >150mg of prednisone then I would recommend using the definition of >150mg in 24h and report the cumulative dose of steroids AND the daily dose in a "mg/Kg/day" format to keep up to the literature.

Results: Overall the results are focused on all patients receiving CS vs those who didn't, and then in CS megadoses vs no megadoses. I believe it would be more useful to change to 3 groups in the tables: No CS, low-dose CS and CS megadoses.

Table 2 is too long, try grouping comorbidities to clinically relevant terms that are well known risk factors for COVID, such as chronic lung disease, chronic cardiovascular disease, renal insufficiency, etc. Table 1 can be integrated at the end of table 2 as it does not add much as a stand-alone table. Revise spelling in the tables.

For table 4 and 5 it would be more useful merged to have the mean/median of lab values instead of the proportion of abnormal labs in both groups as this can be misleading. Table 7 would be more useful if describing megadoses vs regular steroids. Consider adding APACHE, PaFiO2, or other proxies to clinical status upon admission.

There are higher inflammatory markers in patient's receiving megadoses according to tables 4 and 5, this creates an uneven populations to compare the effect of megadoses. There are 2 ways to address this unbalance: inverse weight proportion analysis and propensity-score matching. Consider using one of these to strengthen results.

Mortality metric used in too broad including follow-up of patients already discharged, with a median hospital stay of 12 days, using 30-day mortality would be preferred.

Reviewer #2: From my point of view, it is of little interest to focus the study on the differences between patients who have received megadoses and those who have not. It is much more interesting to find out if the megadoses had negative consequences, in terms of complications, hospital stay or mortality.

Most of the conclusions drawn by the authors are based on bivariate analyzes, which in an observational study like this one, have no more value than merely descriptive. As the authors note, there are a large number of important differences between patients treated with megadoses and those who are not, so the results in terms of complications or mortality are very difficult to interpret. The authors carry out a single multivariate analysis on mortality, in which they have only introduced a few, of the many factors that they have found asymmetrically distributed between groups, without providing an explanation of the selection process carried out. That is, there are enough confounding factors missed in the analysis, so the results most likely have residual confusion. In addition, some important confounding factors that can affect health outcomes in COVID-19 do not seem to have been available, such as saturation and the rest of vital signs, respiratory support, etc. Also, Multivariate analysis, which is essential in this type of observational studies, is limited to mortality results, and has not been done for other interesting results such as, for example, complications of the disease.

Unfortunately, I do not believe that the results are valid for the outcomes of greatest interest, due to the lack of control for relevant confounders, so I cannot recommend the publication in its current state. However, the authors have a valuable database, and their idea of comparing the effect of high doses with other doses of steroids is interesting. I encourage them to choose a relevant outcome, such as mortality, and do a full multivariate analysis, which attempts to elucidate whether mega-doses are associated with higher mortality. They also have an exceedingly large sample, to match patients in whom megadoses were used, with controls with similar clinical characteristics, in which megadoses were not used. In short, they have many possibilities with this database, and I am sure they can make a much more robust analysis than the one presented here, to provide new knowledge about corticosteroids effect in this disease.

6. PLOS authors have the option to publish the peer review history of their article (what does this mean?). If published, this will include your full peer review and any attached files.

Reviewer #1: **Yes: **Edison J. Cano, M.D.

Reviewer #2: **Yes: **Alejandro Rodríguez-Molinero

---

## [Author Response · Author response to Decision Letter 0]

4 Dec 2021

Response to Reviewers:

Reviewers' comments:

Reviewer #1: The authors present their experience with high-dose corticosteroids (CS), referred to as megadoses, and compare these against low-dose CS.

Intro, paragraph 2: consider rephrasing in order to be current with the literature. CS are the only intervention for severe COVID with consistent mortality impact across studies. I would avoid using the references 5-7 that raised controversies in CS use too early in the pandemic and based on theoretical assumptions from non-COVID diseases such as ARDS, SARS or MERS that have distinct etiology, epidemiology, phenotypes and outcomes. Instead, you can highlight even though CS have been widely adopted, there are still questions on dosing, timing and duration of CS that have not been systematically studied at large.

→ We agree with the aforementioned. We have changed it. “Corticosteroids (CS) have been widely adopted, there are still questions on dosing, timing and duration of CS that have not been systematically studied at large.”

Methods: Study conclusion: consider renaming this section Study endpoints to avoid confusion with the use of the word conclusion. The Literature search section does not add to the methods, consider removing.

→ We agree with the aforementioned. We have changed it. 

There are 2 definitions of megadoses, >150 mg of prednisone in 3-4 days and >150 mg of prednisone in 24h. Historically, high-dose CS in ARDS research has been defined as >200mg/Kg/day of methylprednisolone or its equivalent in other CS forms (prednisone, prednisolone, etc). If the authors choose to study >150mg of prednisone then I would recommend using the definition of >150mg in 24h and report the cumulative dose of steroids AND the daily dose in a "mg/Kg/day" format to keep up to the literature.

→ We agree with the aforementioned. We have changed it. “The definition of >150mg in 24h”. About the cumulative dose of steroids AND the daily dose in a "mg/Kg/day", we cannot use this format, because we do not have data to calculate it. 

Results: Overall the results are focused on all patients receiving CS vs those who didn't, and then in CS megadoses vs no megadoses. I believe it would be more useful to change to 3 groups in the tables: No CS, low-dose CS and CS megadoses.

→ We agree with the aforementioned. We have changed it. 

Table 2 is too long, try grouping comorbidities to clinically relevant terms that are well known risk factors for COVID, such as chronic lung disease, chronic cardiovascular disease, renal insufficiency, etc. Table 1 can be integrated at the end of table 2 as it does not add much as a stand-alone table. Revise spelling in the tables.

→ We agree with the aforementioned. We have changed it. 

For table 4 and 5 it would be more useful merged to have the mean/median of lab values instead of the proportion of abnormal labs in both groups as this can be misleading. Table 7 would be more useful if describing megadoses vs regular steroids. Consider adding APACHE, PaFiO2, or other proxies to clinical status upon admission.

→ We agree with the aforementioned. We have changed it. 

There are higher inflammatory markers in patient's receiving megadoses according to tables 4 and 5, this creates an uneven populations to compare the effect of megadoses. There are 2 ways to address this unbalance: inverse weight proportion analysis and propensity-score matching. Consider using one of these to strengthen results.

→ Thank you for your input. We have tried to strengthen the results with multivariate analysis.

Mortality metric used in too broad including follow-up of patients already discharged, with a median hospital stay of 12 days, using 30-day mortality would be preferred.

 → We agree with the aforementioned. We have changed it. 

Reviewer #2: From my point of view, it is of little interest to focus the study on the differences between patients who have received megadoses and those who have not. It is much more interesting to find out if the megadoses had negative consequences, in terms of complications, hospital stay or mortality.

Most of the conclusions drawn by the authors are based on bivariate analyzes, which in an observational study like this one, have no more value than merely descriptive. As the authors note, there are a large number of important differences between patients treated with megadoses and those who are not, so the results in terms of complications or mortality are very difficult to interpret. The authors carry out a single multivariate analysis on mortality, in which they have only introduced a few, of the many factors that they have found asymmetrically distributed between groups, without providing an explanation of the selection process carried out. That is, there are enough confounding factors missed in the analysis, so the results most likely have residual confusion. In addition, some important confounding factors that can affect health outcomes in COVID-19 do not seem to have been available, such as saturation and the rest of vital signs, respiratory support, etc. Also, Multivariate analysis, which is essential in this type of observational studies, is limited to mortality results, and has not been done for other interesting results such as, for example, complications of the disease.

Unfortunately, I do not believe that the results are valid for the outcomes of greatest interest, due to the lack of control for relevant confounders, so I cannot recommend the publication in its current state. However, the authors have a valuable database, and their idea of comparing the effect of high doses with other doses of steroids is interesting. I encourage them to choose a relevant outcome, such as mortality, and do a full multivariate analysis, which attempts to elucidate whether mega-doses are associated with higher mortality. They also have an exceedingly large sample, to match patients in whom megadoses were used, with controls with similar clinical characteristics, in which megadoses were not used. In short, they have many possibilities with this database, and I am sure they can make a much more robust analysis than the one presented here, to provide new knowledge about corticosteroids effect in this disease.

→ We agree with the suggested changes. We have performed the multivariate analysis in mortality and have also focused on the possible adverse effects of megadoses therapy. 

Thank you for all your comments. We hope to have addressed them in the hope of moving the work forward for early publication. 

Kind regards,

Cristina Lavilla

---

## [Editor Report · Decision Letter 1]

9 Dec 2021

Use of Glucocorticoids megadoses in SARS-CoV-2 infection in a spanish registry: SEMI-COVID-19.

PONE-D-21-21800R1

Dear Dr. Lavilla Olleros,

We’re pleased to inform you that your manuscript has been judged scientifically suitable for publication and will be formally accepted for publication once it meets all outstanding technical requirements.

Kind regards,

Aleksandar R. Zivkovic

Academic Editor

PLOS ONE